# Near-universal same-day linkage to ART care among newly diagnosed adults living with HIV: A cross-sectional study from primary health facilities, in urban Malawi

Rachel Chihana[1]*, Chaplain Katumbi[2], Sufia Dadabhai[1], Agness Kaumba[1], Atusaye Mughogho[3], Victor Singano[4], Victor Mwapasa[5], Ken Malisita[6]

1 Johns Hopkins Research Project, Blantyre, Malawi, 2 Itech-Malawi, Lilongwe, Malawi, 3 Ministry of Health, District Health Office, Blantyre, Malawi, 4 Centre for Disease Control, Lilongwe, Malawi, 5 Kamuzu University of Health Sciences, Blantyre, Malawi, 6 Ministry of Health, Queen Elizabeth Central Hospital, Blantyre, Malawi

* rkawalazira@jhp.mw

**Data Availability Statement:** The data can be found in the National Addiction and HIV Data

## Abstract

Malawi HIV treatment guidelines recommend same-day antiretroviral therapy (ART) initiation. Overall 97.9% of Malawians living with HIV (PLHIV) are on ART, same-day ART initiation prevalence and factors that facilitate it have not been fully described. We assessed same-day ART initiation and described individual, health system and health facility infrastructural factors at health facilities supported by expert clients (EC). ECs are lay PLHIV who support other PLHIV. The study was conducted in urban and semi-urban primary health facilities, in Blantyre, Malawi. It was a cross-sectional, descriptive survey of PLHIV and health facility leaders. Eligibility criteria included age ≥ 18 years, new diagnosis of HIV, received counselling from ECs, and offered same-day ART. The study was conducted from December 2018 to June 2021, and 321 study participants enrolled. Mean age (standard deviation) was 33 years (10) with 59% females. In total, 315 (98.1%) initiated same-day ART. Four participants did not because of mental unpreparedness, one wanted to try herbal medicine and one was concerned about stigma related to taking ART. Participants reported health facility accessibility (99%, 318/321), privacy (91%, 292/321) and quality of counselling by EC as excellent (40%, 128/321). Same-day ART was nearly universal. Participants' satisfaction with health services delivery, the presence of EC, and infrastructural characteristics such as adequate privacy were cited as reasons favoring same-day linkage to ART. The most cited reason for not starting same-day ART was mental unpreparedness.

## Introduction

According to United Nations Programme on HIV/AIDS (UNAIDS), in 2020, there were an estimated 25·6 million (68%) people living with HIV (PLHIV) in the Africa region, of a total of 37.7 million people living with HIV globally [1]. In Malawi, HIV prevalence among adults

Archive Program under project ID: NAHDAP-185041.

**Funding:** Research reported in this publication was supported by the Fogarty International Center of the National Institutes of Health, Award Number D43TW0100600. VM received the award. URLS of sponsor's website: https://www.fic.nih.gov/ The funders had no role in study design, data collection and analysis, decision to publish, or preparation of the manuscript.

**Competing interests:** The authors have declared that no competing interests exist.

aged 15–49 years is 8% with the highest (14.2%) occurring in the city of Blantyre, and 97.9% of PLHIV are taking antiretroviral therapy (ART) [2]. A reason for high ART uptake in Malawi is the adoption of the WHO HIV Test and Treat Approach in 2016, which promotes rapid initiation of ART within 7 days of HIV diagnosis [3]. WHO also strongly recommends ART commencement on the same day as HIV diagnosis, after health workers have established the person's willingness and readiness to start ART [4].

Despite the WHO recommendation, there is little literature in Africa describing factors which promote same-day ART initiation. Studies on linkage to care and ART initiation done in Lilongwe, Malawi [5] and South Africa [6] found that health facility factors for non-linkage included long distance to health facility while personal factors included fear of side effects, belief in religious and cultural norms that discourage ART, lack of disclosure to partners and feeling healthy at the time of diagnosis while a study in Kenya showed that health systems factors of lack of adequate time for post-test counseling, poor quality of posttest counselling and testing and poor HIV care coordination were a barrier to linkage as well as health facility factors like access to health facilities and stigma associated with health facility [7]. However, these studies are focused on linkage to care over a period of time. Studies of same-day linkage to care are more rare. One study done in Ethiopia found that health facility factors such as outpatient health centers were more likely to initiate a patient on same-day ART compared to hospitals and personal factors such as patients who were diagnosed their HIV status at the same health facility where they linked to ART, patients with no opportunistic infection and pregnant women were associated with more likelihood of initiating ART on same-day [4]. Same-day initiation has advantages, although studies in sub-Saharan Africa (SSA) have found that same-day ART initiation is low and ranges from 41.9% to 54.2% [4]. In Malawi, data on same-day ART prevalence and its associated factors are limited; one study in 2016 found that same-day ART prevalence was 63% in pregnant women [8]. One strategy used in Malawi to improve time to ART initiation has been to deploy expert clients (ECs) in 29 health facilities in Blantyre district. ECs are lay individuals living with HIV, open to disclosing their HIV status, and with good records of ART adherence. They are trained as volunteer ART counsellors, offering the unique approach of a peer from the same community, and sharing their own experience of living with HIV. ECs talk to other PLHIV on their experience of living positively with HIV, providing them with a better understanding of the disease and less fear of disclosure [9]. At health facilities where ECs are deployed, their roles may include provision of extra post-test counselling, and assistance with navigating and/or escorting patients to the different rooms providing HIV-related services. The EC model provides task shifting from overburdened health workers by supplementing to the posttest counselling done by the HIV testing and counselling providers (HTC). The EC model has been used successfully in Swaziland, Zambia, Botswana, Uganda and South Africa [10] as well as in pilot studies in Balaka and Machinga, rural districts in southern region of Malawi [9]. Malawi has achieved the second of the UNAIDS goals to have 95% of those living with HIV on sustained ART by 2030, as 97.9% of PLHIV are on ART [2] providing evidence that after diagnosis, most people do eventually initiate ART, even if it may not be on the same day of diagnosis. There are about 2.1% of PLHIV who are not on treatment, posing a risk of morbidity and mortality in these individuals and HIV transmission to others.

There is need to investigate same-day ART start in Malawi and identify factors, including EC, which may be associated with same-day linkage to ART care. The understanding of these determinants will allow targeted improvements in this key metric, and will improve health system efficiency to allow health facilities to continue to serve high numbers of clients [11]. Our study sought to assess same-day ART linkage prevalence, and describe individual, health facility, and systems factors influencing same-day linkage to ART care among PLHIV in primary health facilities supported by EC.

## Methods

This was a quantitative descriptive cross sectional study, conducted at Limbe and South Lunzu health facilities within Blantyre district in southern Malawi. Blantyre district covers an area of 220 square kilometres and has a population of 932,000 [12]. Blantyre is Malawi's second largest city and the largest commercial and industrial city in the country [12]. This was a pilot study and these facilities were selected for convenience because they are busy facilities and hence the sample size was easily achieved. South Lunzu (semi-urban) initiates about 35 people per month on ART, and Limbe (urban) primary health facilities provides care to an average of 100 newly diagnosed PLHIV per month. The facilities have HIV testing and counselling (HTC) clinics where the following services are provided: (1) group counselling (2) pre-test counselling (3) testing and (4) posttest counselling. All these services are conducted in a separate counselling room, by an HTC provider. After HIV diagnosis, they see the EC for peer counseling and then the ART provider. Participants were approached for the study after their session with the EC. All HIV newly diagnosed adults from South Lunzu and Limbe health facilities were approached for screening. Additional inclusion criteria were: adults of over 18 years of age, newly diagnosed of HIV, provided with extra counselling by an EC and willing to provide written informed consent. Additional health systems data were collected from health care workers (HCW) who were leaders of each health facility.

### Sample size

We used a single population formula to calculate the number of participants needed to determine prevalence of same-day ART acceptance. We assumed same-day ART initiation would be 63% based on the previous Malawian study [8]. The calculated sample size was 298 participants. By adding a 10% non-response rate, the sample size was arrived at 328.

### Data collection, management and analysis

Two study staff at each facility recruited participants and conducted interviews with participants after their session with the EC. Those willing to start same-day ART then proceeded to the ART provider room, while those unwilling left for their homes. Participants filled a structured questionnaire (S1 Questionnaire) with comprehensive socio-demographic information. Participants' experience at the health center, including waiting time, and medication stock outs, was also assessed as well as the health infrastructural characteristics which included accessibility and privacy. A checklist (S1 Checklist) administered to each leader at both health facilities collected additional information on health systems building blocks of health service delivery, health work force and also access to essential medicines. These data collection tools were put together specifically for this study and were piloted at a different health facility before the study commenced. This led to revision of some questions in the questionnaire before they were implemented in the study. The main outcome variable was same-day ART initiation, with categorical response of (Yes, No), verified by staff or health book. Dependent variables collected included (1) personal variables, which included sociodemographic data of study participants, (2) health facility infrastructural variables, which included accessibility of health facilities, privacy and distance between provider rooms, and (3) health systems which included variables in the health systems building blocks of health service delivery, health work force and access to essential medicines. Questionnaires and checklists were filled out using Open Data Kit (ODK)and imported into STATA version 14.2 for analysis. Using STATA, descriptive data were summarized using proportions and means of the whole study sample, those linked to ART and those not linked to ART where appropriate.

### Ethics

Ethical approval to conduct the study was obtained from Kamuzu University of Health Sciences, College of Medicine Ethics Committee (COMREC). All study participants gave written informed consent in Chichewa language.

## Results

The study was conducted from December 2018 to June 2021. A total of 333 newly-diagnosed PLHIV were approached to take part in the study, and 321 (96%) accepted study participation, including 161 participants from Limbe and 160 from South Lunzu. More participants from Limbe declined participation (4.9%, 8/161) compared to South Lunzu (2.5%, 4/160). Reasons for declining included not interested (8/12), in a hurry (3/12) and wanted husband's consent (1/12).

### Characteristics of study participants

The baseline characteristics of the participants are shown in Table 1. The mean age (standard deviation [SD]) of participants was 33 years (10), most were female (59%, 189/321) and the majority had a source of income (67%, 215/321). Study participants reported taking the HIV test on a voluntary basis (66%, 211/321) as opposed to provider initiated, and most accepted their HIV results (99%, 318/321) and knew someone on ART (77%, 246/321). A few study participants reported fear of stigma (5%, 17/321) related to disclosure of HIV status.

### Linkage to ART

There were 315 out of 321 participants who initiated ART on the same-day of HIV diagnosis, representing a total linkage point prevalence of 98.1%. Prevalence of same-day linkage for Limbe health facility was 97.5% (157/161) while that of South Lunzu was 98.75% (158/160). There were 6 people who were not linked to ART for the following reasons: 67% (4/6) were mentally not ready to start ART immediately, 17% (1/6) wanted to use herbal medicine and another 17% (1/6) were afraid of stigma related to taking ART. The mean age (SD) of those not linked to ART did not differ from those who did accept ART: 32 years (7) and they were predominantly female (67%, 4/6).

### Responses from health care workers and clients regarding health systems performance

Health facility leaders reported that Limbe had a total of 18 HTCs, 35 ART providers and 10 ECs who worked 8 hours a day while South Lunzu had 9, 31 and 7, respectively. On a single day, at Limbe health facility, there are on average 10 HTC counsellors, 10 ART providers and 10 ECs, while at South Lunzu health facility, there are 3 HTC providers, 4 ART providers and 7 ECs providing services. For both health facilities, there was no single day in the first quarter of 2020 that HTC, ART and ECs services were not available. The participants rated the counselling services provided by, and the quality of interaction between themselves and HCWs and ECs (Table 2). The majority (74%, 238/321) of participants indicated that they waited for less than 1 hour before they received services at the HTC clinic. ECs made contact with almost all persons, with 97% (311/321) of participants indicating that they had received assistance from EC, with directions and/or escort (navigation assistance) to different rooms for services. Those who wanted to continue ART at a different facility were still able to initiate same-day ART, with an established referral system of patients who get diagnosed at one facility but would like

**Table 1. Descriptive socio demographic characteristics of persons newly diagnosed with HIV at primary health facilities in Blantyre, Malawi.**

| Sociodemographic characteristic (N = 321) | n (%) or mean (SD) |
|---|---|
| Age in years, mean (SD) | 33 (10) |
| Gender | |
| female | 189 (59) |
| male | 132 (41) |
| Source of lighting | |
| electricity | 177 (55) |
| lamp/candle | 144 (45) |
| none | 0 (0) |
| Source of water | |
| tap | 22 (69) |
| borehole | 96 (30) |
| river/stream | 4 (1) |
| Cooking mode | |
| electricity | 26 (8) |
| charcoal | 250 (78) |
| firewood | 45 (14) |
| Sanitary facility | |
| flush | 35 (11) |
| pit latrine | 286 (89) |
| Mode of transport | |
| walked | 71 (22) |
| public transport | 244 (76) |
| private transport | 6 (2) |
| Smoking: | |
| yes | 29 (09) |
| no | 292 (91) |
| Alcohol: | |
| yes | 64 (20) |
| no | 257 (80) |
| Education status | |
| no formal education | 23 (7) |
| Primary school level | 156 (49) |
| Secondary school level | 125 (39) |
| Tertiary level | 17 (5) |
| Religion | |
| Christian | 285 (89) |
| non-Christian | 36 (11) |
| Location from facility | |
| <5km | 202 (63) |
| >5km | 119 (37) |
| Distance in KM*, mean (SD)** | 4.3 (1.9) |
| Number of sexual partners | |
| no partner | 26 (8) |
| 1 partner | 207(65) |
| >1 partner | 88 (27) |

*KM:—Kilometers

**SD:—Standard deviation

**Table 2. Rating by study participants of counselling services provided by HCWs and of quality of interaction between the study participants and HCWs.**

| Score, n (%) | Excellent | Good | Average | Bad | Total |
|---|---|---|---|---|---|
| Quality of counselling by EC* | 127 (39.56) | 191 (59.50) | 2 (0.62) | 1 (0.31) | 321 (100) |
| Quality of counselling by HTC** providers | 112 (34.89) | 207 (64.49) | 2 (0.62) | 0 (0.00) | (321) 100 |
| Quality of interaction with EC | 134 (41.74) | 185 (57.63) | 2 (0.62) | 0 (0.00) | (321) 100 |
| Quality of interaction with HTC providers | 125 (38.94) | 194 (60.44) | 2 (0.62) | 0 (0.00) | (321) 100 |

*EC:—Expert client

**HTC:—HIV testing and counselling

to receive their ART from another. The health care workers also agreed that in the first quarter of 2020 there were no stock outs of HIV testing materials or ART drugs.

## Description of health facility infrastructural characteristics by study participants

Study participants described infrastructural characteristics (Table 3) which included: health facility accessibility, privacy in accessing ART services and conduciveness of the distances and location of some rooms where HTC services were provided. Most participants found the health facility accessible (99%, 318/321) and the distance between ART room and HTC clinic conducive (85%, 274/321).

## Discussion

In this study, we found that same-day linkage to ART of newly diagnosed PLHIV was very high (98.1%). We also found that most participants had positive views of the service delivery and infrastructure at the facilities, which may have facilitated same-day linkage to care. The highly rated services described by study participants included counselling services, clinic navigation and conducive and accessible health facilities. Health facility leaders described adequately staffed facilities without ART and medical supplies stock outs. In total, participants and facility leaders described a well-functional infrastructure which promoted and enabled newly diagnosed persons to accept same-day ART initiation.

This high same-day ART initiation prevalence found in our study is in sharp contrast with findings from previous studies in sub-Saharan Africa (SSA) which found same-day linkage ranging from 41.9%- 54.2% [4]. Prior barriers to same-day initiation found in previous studies included the need for newly diagnosed PLHIV to start ART at a different facility from where they tested [4]. Studies conducted elsewhere in SSA have shown that participants who are

**Table 3. Responses of study participants about their experience of health facility infrastructure.**

| | Yes n (%) | No n (%) | Total n (%) |
|---|---|---|---|
| **Health facility characteristics** | | | |
| 1. Was the health facility accessible to you? | 318 (99.) | 3 (1.) | 321 (100) |
| 2. Was there privacy in accessing ART? | 292 (91.25) | 28 (8.75) | 321 (100) |
| 3. Did you find distance between ART room and HTC clinic conducive? | 274 (85.36) | 47 (14.64) | 321 (100) |
| 4. Did you find location of EC in the HTC room conducive? | 306 (95.33) | 15 (4.67) | 321 (100) |

ART:—Antiretroviral therapy

HTC:—HIV testing and counselling

EC:—Expert client

tested in a health facility in which they are linked to ART services are two times more likely to initiate same-day ART compared with those tested from another health facility or in a community HIV testing setting and subsequently referred [4, 10, 13]. In situations where newly diagnosed PLHIV request referral to another health facility for supply of their ART, Limbe and South Lunzu health facilities have an existent referral system whereby they commence the patient on ART first and refer to another health facility for continuation of supply of ART, which may have contributed to the high linkage. Another factor contributing to higher same-day initiation is that this study setting was only primary health facilities, while previous studies included primary and secondary health facilities [4]. Patients who present to primary health-care facilities are more likely to have uncomplicated clinical presentation, making it easier for health workers to commence them on same-day ART, unlike in a secondary health facility, where health care workers may have to conduct other investigations or stabilize the patient before ART is commenced [4]. This is also in line with other studies that found that patients without opportunistic infections were two times more likely to initiate same-day ART compared with patients with one or more opportunistic infections [4]. Similarly, a study in South Africa reported that patients who presented with less advanced clinical disease were more likely to accept same-day ART initiation [14]. However, such findings are in contrast with several qualitative studies conducted in Malawi and other SSA countries which reported the absence of symptoms or signs of ill health as a major reason for deferring same-day ART [8].

The finding that most participants had positive views about the health facility is consistent with findings from previous studies conducted in Malawi and other countries in SSA which found that patients' satisfaction with the quality of services, patient-health worker relationships, time efficiency at the facility, good coordination of the different services at the HTC clinic, availability of staff and supplies, and accessibility of health facilities all contribute to reducing barriers to ART initiation [8, 12–15]. Our study further revealed that the majority of study participants were content with the privacy of the facility and especially the proximity of the HTC, EC and ART rooms. Unfavorable distance between these rooms can instill fear of inadvertent disclosure of HIV status and therefore have an impact on ART linkage.

The finding that four out of six participants who were not linked to ART did so because they were mentally not ready is consisted with findings of a study conducted in eastern and southern parts of Africa, which highlighted the fact that the process of HIV status acceptance takes time [10]. The 'test and treat' approach has significantly removed the psychological component where one is required time to digest the news of their new HIV status and make their decision on ART [10]. The participants who were not linked to ART on the day of diagnosis might have initiated ART later, but this was outside the mandate of our study.

Our facilities were supported with lay health workers (EC) and we found that they were a critical part of the health facility and may have contributed to the high same -day linkage through the support they provided to clients. Task-shifting to meet human resources for health facilities is recommended by WHO [15]. Evidence encourages such task-shifting in HCT settings in LMIC in SSA that are affected by the HIV epidemic, and have a critical shortage of skilled health care providers which provides additional efficiencies to the health services [16]. Having additional personnel to provide both emotional support and navigation services contributes to the smooth functioning of the health facility and contributes to supporting PLHIV on their day of diagnosis.

Our study provides insight on multi-dimensional factors associated with same-day ART linkage in primary health facilities, but it had limitations. This study did not explore community factors like family or community support structures that might affect same-day ART initiation. The study participants described their facility experience, which could have been a source of social desirability bias. Linkage outcome, the perception of the health facility and

health systems and personal characteristics of the 12 adults living with HIV who refused study participation is not known. It is important to note that there is potential for selection bias if the linkage outcome of these 12 individuals and their perceptions are similar. Our findings show that in an under-resourced lower income country, extremely high rates of same-day ART start can be achieved at the primary health facility setting. We suggest that participants' satisfaction with health services delivery and infrastructural characteristics may have favored linkage to ART of newly diagnosed PLHIV, and that primary health facilities supported by lay health workers can provide high patient satisfaction. Our findings also suggest that mental unpreparedness may contribute to unsuccessful linkage to ART. We recommend that health facilities be supported with human resources, medical drugs and supplies and appropriately designed infrastructure in order to deliver services optimally. Health facilities must be accessible and be designed in a manner that encourages privacy. Future directions could include research to determine what type of counselling is most effective in encouraging newly diagnosed patients to start ART on the same-day as diagnosis.

## Supporting information

**S1 Checklist. A checklist used for data collection in the study.**
(DOCX)

**S1 Questionnaire. A structured questionnaire used for data collection in the study.**
(DOCX)

## Acknowledgments

The authors would like to thank the study participants as well as the study team for their significant contributions to the study. We thank Alison Roxby MD at the HIV Vaccine Trials Network for editorial assistance. We also thank Dr Bonus Makanani for his contribution in the implementation of this research.

## Author Contributions

**Conceptualization:** Rachel Chihana, Sufia Dadabhai, Atusaye Mughogho, Victor Singano, Victor Mwapasa, Ken Malisita.

**Data curation:** Chaplain Katumbi, Agness Kaumba.

**Formal analysis:** Rachel Chihana, Chaplain Katumbi.

**Methodology:** Rachel Chihana.

**Project administration:** Rachel Chihana, Sufia Dadabhai, Agness Kaumba.

**Supervision:** Rachel Chihana, Victor Mwapasa.

**Writing – original draft:** Rachel Chihana, Sufia Dadabhai.

**Writing – review & editing:** Sufia Dadabhai, Agness Kaumba, Atusaye Mughogho, Victor Singano, Victor Mwapasa, Ken Malisita.

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
