## [Decision Letter · Decision Letter 0]

6 Jan 2023

PGPH-D-22-01910

Near-universal same day linkage to ART care among newly diagnosed adults living with HIV: A cross sectional study from primary health facilities, in urban Malawi

Dear Dr. Chihana,

Thank you for submitting your manuscript to PLOS Global Public Health. After careful consideration, we feel that it has merit but does not fully meet PLOS Global Public Health’s publication criteria as it currently stands. Therefore, we invite you to submit a revised version of the manuscript that addresses the points raised during the review process.

We look forward to receiving your revised manuscript.

Kind regards,

Ari Probandari, PhD

Academic Editor

Journal Requirements:

1. Please send a completed 'Competing Interests' statement, including any COIs declared by your co-authors. If you have no competing interests to declare, please state "The authors have declared that no competing interests exist". 

3. Please amend your Data Availability Statement and indicate where the data may be found

Reviewers' comments:

Reviewer's Responses to Questions

**Comments to the Author**

1. Does this manuscript meet PLOS Global Public Health’s publication criteria? Is the manuscript technically sound, and do the data support the conclusions? The manuscript must describe methodologically and ethically rigorous research with conclusions that are appropriately drawn based on the data presented.

Reviewer #1: Partly

Reviewer #2: Yes

2. Has the statistical analysis been performed appropriately and rigorously?

Reviewer #1: No

Reviewer #2: Yes

3. Have the authors made all data underlying the findings in their manuscript fully available (please refer to the Data Availability Statement at the start of the manuscript PDF file)?

Reviewer #1: Yes

Reviewer #2: Yes

4. Is the manuscript presented in an intelligible fashion and written in standard English?

Reviewer #1: Yes

Reviewer #2: No

5. Review Comments to the Author

Reviewer #1: Thank you so much for the opportunity to review this paper. This paper explored an important topic of the same day linkage to ART initiation that is a recommended strategy to achieve epidemic control of HIV infection. The context of Malawi, with a high coverage of PLHIV on ART as well as high same day ART initiation, compared to other SSA countries (as described in the Introduction), is an interesting perspective, especially to understand what the individual-, facility is- and system-level factors that influence this high coverage. However, my overall comment is this paper lack a specific framework of assessing the uptake of same day linkage to ART and methodological details of how authors approach the primary aim of this study. Details of my feedback are as below:

Title

1. It is not clear whether the working title is one at the page 1 or at the page 3 (Line 49 and 50)

Introduction

1. Introduction could be structure better with each paragraph consistently summarizing one main idea, for example, the first paragraph could focus on epidemiology profile globally, regionally and specifically with Malawi. It could follow with some policy context around these numbers. (e.g., it is unclear what are data that limited in line 61, and the goal of 95-95-95 that had not been introduced properly in the line 75 page 4).

2. The next (third paragraph) could focus on any past studies on the factors (the individual-, facility is- and system-level factors) that influences uptake of same-day linkage to ART initiation in Malawi and SSA. Currently it is a little bit lacking, with more focused are given only to the availability of EC (expert clients)

3. The knowledge gap and the primary aim of this paper are a bit vague in the last paragraph of this introduction. I will suggest authors to also consistently use the terms throughout the paper (e.g., health care centre or health facility)

Methods

1. As I mentioned in the overall comment, this paper lacks a framework to describe the different level of individual-, health facility, and system- level factors that influence the uptake of same-day ART initiation.

2. Sample size: I do not understand on the justification of the selection of Blantyre district (other than what been mentioned in Introduction about higher coverage of same-day ART), the context of this district, the size and justification of selection of only two facilities (south Lunzu and Limbe) and if there is randomization process taken place in this selection (or all are selected purposefully and how this maybe create bias).

3. There is no detailed explanation about variables (other than the main outcome) – such as predictors, potential confounders or other type of variables.

4. Data sources are not clearly defined and questionnaire that been used in this study are not attached. While also there is no explanation how the questionnaire is made, and whether it has been piloted before.

5. This paper will very be benefitting with a statistical method that can check any association between main outcome and variables of interest. At this current state, the analysis seems lacking.

Results

1. Inconsistent way to present the results, line 132 page 6, also line 152 – 153

2. Table 1. Source of water – tap, the number is mistype

3. Line 155, health facility leaders are not defined (or explained) before

4. The terminology “views of health care workers and clients” in the page 153 seems misleading since it seems like a qualitative study instead.

Reviewer #2: Article well-written.

1. Topic: Topic followed scientific format

2. Abstract: Abstract well written

3. Introduction section

a. In line 52, author did not define UNAIDS at the initial, knowing that USAID is an acronym.

b. In line 53, 54 and 55, Author can reverse by joining a two sentence to a singular one for example; In Malawi, HIV prevalence among adults aged 15- 49 years is

8%, and 97.9% of PLHIV are taking antiretroviral therapy (ART) with the highest ((14.2%) occurring in Blantyre (2).

4. Methods: Methods described the scientific approach the manuscript.

5. Results: Results were well written however, the manner of table presentation should comply with the scientific format of presenting results in a tabular form. For example, table should contain 3 lines; top, middle and the bottom lines.

6. PLOS authors have the option to publish the peer review history of their article (what does this mean?). If published, this will include your full peer review and any attached files.

**Do you want your identity to be public for this peer review?** For information about this choice, including consent withdrawal, please see our Privacy Policy.

Reviewer #1: No

Reviewer #2: **Yes: **Abbas Abel Anzaku

---

## [Editor Report · Decision Letter 1]

18 May 2023

Near-universal same day linkage to ART care among newly diagnosed adults living with HIV: A cross sectional study from primary health facilities, in urban Malawi

PGPH-D-22-01910R1

Dear Dr Chihana,

We are pleased to inform you that your manuscript 'Near-universal same day linkage to ART care among newly diagnosed adults living with HIV: A cross sectional study from primary health facilities, in urban Malawi' has been provisionally accepted for publication in PLOS Global Public Health.

Best regards,

Ari Probandari, PhD

Academic Editor

The authors have provided sufficient responses to the reviews adequately.